# Jumping in the Chiral Pool: Asymmetric Hydroaminations with Early Metals

**DOI:** 10.3390/molecules28062702

**Published:** 2023-03-16

**Authors:** Sebastian Notz, Sebastian Scharf, Heinrich Lang

**Affiliations:** TU Chemnitz, Research Center for Materials, Architectures and Integration of Nanomembranes (MAIN), Research Group Organometallic Chemistry, Rosenbergstraße 6, D-09126 Chemnitz, Germany

**Keywords:** chiral pool, early metals, hydroamination, homogeneous catalysis, stereoselectivity, regioselectivity

## Abstract

The application of early-metal-based catalysts featuring natural chiral pool motifs, such as amino acids, terpenes and alkaloids, in hydroamination reactions is discussed and compared to those beyond the chiral pool. In particular, alkaline (Li), alkaline earth (Mg, Ca), rare earth (Y, La, Nd, Sm, Lu), group IV (Ti, Zr, Hf) metal-, and tantalum-based catalytic systems are described, which in recent years improved considerably and have become more practical in their usability. Additional emphasis is directed towards their catalytic performance including yields and regio- as well as stereoselectivity in comparison with the group IV and V transition metals and more widely used rare earth metal-based catalysts.

## 1. Introduction

The coupling of carbon and nitrogen bonds is of great importance to organic chemistry [1,2]. The thusly formed nitrogen-containing compounds including *N*-heterocycles offer diverse applications not only in material sciences, but also in natural product synthesis and pharmaceutical chemistry. One synthetic concept in the mostly applied preparation of such molecules is the hydroamination reaction [3,4,5,6,7,8,9,10,11].

Hydroamination is the addition of an N–H bond of a primary or secondary amine across a carbon–carbon double or triple bond of, for example, alkenes, alkynes, dienes or allenes, resulting in an optimal atomic economy of 100% [5,6,12,13]. However, asymmetric *C,N* coupling processes including the Aza–Wacker [14], Buchwald–Hartwig [14], aminoacetoxylation [14] and photoredox (aminium radicals) [15] reactions are less atomically efficient than hydroaminations. Depending on the substrates used, hydroamination reactions occur either intermolecularly, in which the relevant functional groups are part of the separated starting materials, or intramolecularly, wherein the substrates combine both the amine and unsaturated C=C or C≡C building blocks in a single molecule [5].

Commonly, the intermolecular hydroamination of alkenes and alkynes results in the formation of Markovnikov and/or anti-Markovnikov regioisomers [5]. In the case of allenes and alkynes, *E/Z* isomers are produced [6,7,16]. Intramolecular hydroamination favors the Markovnikov product giving *α*-alkyl *N*-heterocycles for alkene substrates [5]. In addition, substituents in the *β*-position to the amino unit of the nitrogen-bonded unsaturated organic carbon–hydrogen substrate affect the reaction rate, which is known as Thorpe–Ingold effect [17].

Hydroamination reactions are thermodynamically neutral [5,6,18,19]. Due to the electrostatic repulsion between the nitrogen lone pair, the *C,C π*-system and the orbital symmetry-forbidden [2 + 2] cycloaddition, hydroamination reactions possess a high reaction barrier despite being kinetically favored as caused by the increase in the total bonds. Therefore it is necessary to catalyze or run the respective reactions at a high temperature [5,6].

To the best of our knowledge, the first hydroamination in solution, the *C,N* coupling reaction of *p*-toluidine with cyclohexene, was reported by Hickinbottom in 1932 [20]. Shortly after this, Kozlov et al., published the catalytically controlled hydroamination of an amine with an alkyne in the presence of mercury (II), copper (II) or silver (I) halides as catalysts [21,22]. In 1971, Coulson described the reaction of amines with alkenes by using catalytic active Rh and Ir species [23]. Since then, the field of *C,N* coupling via hydroamination has been expanding [6,12,24,25,26,27,28,29,30]. Early transition metals of group IV and V from the periodic table of elements were introduced by Bergman and Livinghouse [31,32]. Rare earth metal-based catalysts were launched by Marks dating back to 1989 [33], and early main-group elements as catalytic systems were established at the start of the new millennium [34].

Main-group or lanthanide-element-based catalysts are generally less tolerant towards amines and alkenes featuring polar functional groups, e.g., esters, ketones and alcohols and are more sensitive towards air and moisture as compared to late transition metal complexes. However, they exhibit an overall higher reactivity, a better regioselectivity and are more ecologically friendly compared with late transition metal hydroamination catalysts. Hence, early-metal-based catalysts are the preferred catalysts over expensive and often toxic late transition metal ones, especially in intramolecular hydroamination reactions [5,6,13].

Since intermolecular alkene hydroaminations and intramolecular cyclization reactions of aminoalkenes may form stereocenters, chiral catalysts are required to obtain enantiopure isomers. For this to occur, ligands such as 1,1′-bi-2-naphthol(=BINOL), 2,2′-diamino-1,1′-binaphthaline(=DABN) or 2,2′-bis(diphenylphosphino)-1,1′-binaphthyl(=BINAP) derivatives are best suited, due to their bulk and (excellent) enantioselectivity [35,36]. In addition to these synthetic, accessible but hard to purify and difficult to up-scale biaryls, a series of enantiopure building blocks provided by nature are also of great benefit. The so called chiral pool-based ligands are readily available, ecologically friendly and hence “green” [37,38,39,40]. Due to their low cost, high abundance and general sustainability, the chiral pool has been extensively utilized by synthetic chemists in the preparation of ligand systems in enantioselective catalysis of natural products as well as pharmaceutical agents, with an extensive literature available on these topics [39,41,42,43,44,45,46]. The most relevant chiral pool motifs in hydroamination reactions are amino acids, both proteinogenic and non-proteinogenic, alkaloids and terpenes (Figure 1).

While originally only naturally occurring, enantiopure compounds were considered to be part of the chiral pool, modern definitions tend to include on a significant scale industrially produced, enantiomerically pure compounds, which can be obtained either by racemate cleavage, enantioselective synthesis or the derivatization of enantiomerically pure natural products [47].

Herein, we focus on asymmetric homogeneous metal-catalyzed hydroamination coupling reactions using early group IV and V transition, rare earth and main-group metals as catalysts featuring ligands originating from the chiral pool. The regio- and stereoselectivities, activities, conversions and yields towards the formation of the corresponding hydroamination products will be discussed in dependence of the metals, chiral pool motifs and the appropriate catalysis conditions.

## 2. Chiral Pool-Based Catalysts for Asymmetric Hydroamination Reactions

In the first catalytic hydroamination reaction dating back to the early nineteen thirties, group XI and XII metal halides were applied as catalysts [21,22]. Since then, a multitude of metal compounds have been researched for their suitability as catalytic active systems in inter- and intramolecular hydroamination reactions [3,5,48]. The catalysts can be differentiated into late and early transition metals, rare earth metals and early main-group elements. In the following, the application of hydroamination catalysts especially featuring ligands originating from the natural chiral pool will be discussed in detail and compared to non-chiral pool ligands.

### 2.1. Late Transition Metals

Late transition metal catalysts containing, for example, neutral chiral (di)phosphine-, bipyridine- or bisoxazoline-based ligands to induce regioselectivity and chirality have recently been used [14,49,50,51]. In addition, chiral pool relevant motifs such as *α*-hydroxy acids and sugar acids including tartaric acid were introduced as chiral centers in phosphine ligands. Examples include 2,3-*O*-isopropyliden2,3dihydroxy-1,4-bis(diphenyl-phosphino)butane(=DIOP) or (2*S*,3*S*)-(−)-bis(diphenylphosphino)butane(=CHIRAPHOS). However, these systems induce generally lower *ee* values in comparison to non-chiral pool-derived phosphines, e.g., BINAP derivatives. Detailed discussions on this topic can be found elsewhere [3,7,14,16,49,50,51,52]. During the last three decades, focus has also been directed to hydroaminations applying early metal catalysts, including those featuring chiral-pool-derived ligand peripheries, especially for their application in intramolecular hydroaminations [4,5,6].

### 2.2. Early Transition Metals

Early transition metals and rare earth metals have been extensively studied in intramolecular hydroamination catalysis [24]. The HSAB principle states that “hard metals” bind to “hard ligands”. Therefore, early metals have been combined with ligands, such as amines, alcohols and ethers. For transition metals of group IV of the periodic table of elements and rare earth metals, cyclopentadienyls have also been proven to be excellent ligands for the catalytic active center [33,53,54,55,56,57,58].

#### 2.2.1. Rare Earth Metals

The intramolecular hydroamination of non-activated olefins using rare-earth metal complexes was pioneered by the group of Marks in the 1990s [33,54]. The majority of the catalysts featuring cyclopentadienyl entities allows the efficient generation of racemic or enantio-enriched *N*-heterocycles [33,54,59,60,61,62,63]. Using these systems, mechanistic studies were undertaken and two mechanisms were proposed [64,65,66,67]. The *σ*-insertion mechanism suggested by Marks et al., (Figure 2a) postulates a rapid, reversible migratory olefin insertion of the metal amide followed by a slower, irreversible rate-determining metal alkyl bond protonolysis by a further substrate molecule [29,53,54,55,61,68,69,70]. The turnover-limiting M–C *σ*-bond aminolysis occurs by a substrate molecule which is followed by a kinetically favored displacement of the *N*-heterocycle, as confirmed by deuterium-labelling experiments [29,54,61,68,69]. Exemplary NH/ND kinetic isotope effect (=KIE) and isotopic perturbation studies on Cp*_2_LnR (Ln = La, Nd, Sm, Y, Lu; R = H, CH(TMS)_2_, *ƞ*^3^-C_3_H_5_, N(TMS)_2_; TMS = SiMe_3_) complexes were carried out to define the stereochemistry of the corresponding heterocycles [71]. These studies found that the NH/ND KIE cannot be derived from protonolysis of a previously formed Ln–C bond. In order to explain this finding a non-insertive catalytic cycle was proposed (Figure 2b), involving a second coordinated amine substrate, partially transferring one of its two NH protons to the terminal alkene carbon atom to form the pyrrolidine product by insertion (Figure 2b) [29,68,71]. Finally, the coordinated pyrrolidine is released by a new substrate molecule [29,68,71]. The two discussed mechanisms for intramolecular hydroamination reactions using rare earth (or main-group) metal-based catalysts compete with each other [71].

One example is the enantioselective and regioselective hydroamination of aminoalkenes **1a**,**b** and **2a**,**b** to give chiral pyrrolidines **3a**,**b** or piperidines **4a**,**b** with *C*_1_-symmetric lanthanide *ansa* complexes: (*S*)-[Me_2_Si(*η*^5^-C_5_Me_4_)(*η*^5^-C_5_H_3_R*)]Ln-E(TMS)_2_ (**5a**–**n**, Ln = Y, La, Sm, Nd, Lu; E = N, CH; R* = (−)-menthyl, (+)-neomenthyl; TMS = SiMe_3_) [54,73] and (*S*)-[Me_2_Si(OHF)(*η*^5^-C_5_H_3_R*)]LnN(TMS)_2_ (**6a**–**c**, Ln = Y, Sm, Lu; OHF = *η*^5^-octahydrofluorenyl) [60] serving as catalysts (Figure 1, Table 1) [54,60,73].

From Table 1 it can be seen that catalysts **5a**–**n** and **6a**–**c** achieve moderate to high *ee* values despite facile epimerization under the catalytic reaction conditions, due to reversible protolytic cleavage of the metal cyclopentadienyl bond [54,60,61,73,74]. A further characteristic is that the (+)-neomenthyl-containing catalysts **5c**–**g**,**j**,**k** (Table 1, entries 3–12, 21–27) form the corresponding (*R*)-(−) enantiomers, while the (−)-menthyl comprising derivatives **5a**,**b**,**h**,**i** (Table 1, entries 1, 2, 13–19) and **6a**–**c** (Table entries 32–46) give the respective (*S*)-(+)-configured *N*-heterocycles **3a**,**b** and **4a**,**b**, with the exception of entries 18 and 44 in Table 1. The chirality of the catalysts has no effect on the optical rotation of the product. However, when lutetium complexes **5l**–**n** (Table 1, entries 28–31) are used as catalysts, then aminopentenes **1a**,**b** are cyclized to give enantiomers of **3a**,**b** in the exact opposite enantioselectivity to **5a**–**k** (M = Y, La, Nd, Sm) [54,73]. The best overall enantioselectivities with up to 74% *ee* were obtained for **1b** using the (−)-menthyl-substituted samarium complexes (*S*)-**5h**,**i** at −30 °C (Table 1, entry 17) [54,60,73]. Generally, both (*R*)- (Table 1, entries 19 and 21) and (*S*)-enantiomers (Table 1, entries 13 and 20) of **5h**,**j** show comparable *ee* values (**5h**: 60 vs. 62%; **5j**: 52 vs. 55%) and the same optical rotation for pyrrolidine **3a** at 25 °C [54,60,61,73]. For catalysts **6a**–**c** using **1a**,**b** and **2a**,**b** as substrates, an *ee* value as high as 67% ((*S*)-**6a**) was obtained (Table 1, entry 36). The activities of **6a**–**c** are in general lower than those of **5a**–**n,** hence, the use of catalysts **6a**–**c** requires higher temperatures [60].

Additionally, catalysts (*S*)-**5h** and (*S*)-**6a**,**b** were studied in the hydroamination/cyclization of sterically hindered aminoalkenes *E*-**7a**,**b** and *Z-***8**, producing pyrrolidines **9a**,**b** and piperidine **10** with good to excellent yields and *ee* values as high as 68% (with (*S*)-**6a** as a catalyst (Table 2, entries 6 and 7), albeit at much harsher conditions (Table 2) [70]. Generally, the (+)-enantiomers **9a**,**b** and **10** are formed. However, using (*S*)-**6a** as pre-catalyst for the cyclization of aminohexene *Z-***8** to piperidine **10**, the corresponding (-)-enantiomer was obtained.

Substrate screening was later extended by the group of Marks et al., towards the conjugated 1,3-aminodienes **11** and **12a**,**b** using (*S*)-**5a**,**b**, (*S*)-**5h** and (*S*)-**6b** as organolanthanide catalysts (Table 3) [61]. The reaction rate is higher for the aminodienes **11** and **12a**,**b** than for the corresponding aminoalkenes **1a** and **2a**,**b**, despite increased steric hindrance of the cyclization transition state [25,61]. However, the enantioselectivity is generally lower, with the exception of the formation of *N*-heterocycle **14a** with (*S*)-**6b** as a catalyst showing up to 71% *ee* (Table 3, entry 10) [25,61]. The authors also show the high stereoselectivity of the intramolecularly proceeding aminodiene hydroamination by concise synthesis of naturally occurring alkaloids (±)-pinidine and (+)-coniine from easily accessible diene substrates [25,61].

In 2003, Marks et al., published a series of *C*_2_-symmetric bis(oxazolinato)lanthanum complexes and discussed their use as efficient catalysts for the intramolecular hydroamination of aminoalkenes and aminodienes [75]. Two complexes out of the reported series possess L-valinol- (**15a**) (Table 4, entry 1) and L-*tert-*leucinol-derived (**15b**) (Table 4, entry 2) chiral pool ligands for the cyclization of **1b** (Figure 2) [75]. However, the observed enantioselectivities were with 6% (**15a**) and 39% (**15b**) at 25 °C lower than those for the non-chiral-pool-based systems with aryl functionalities in the *α*-position to the nitrogen atom, which result in up to 67% *ee* for substrate **1b**. Generally, it can be stated that lanthanides possessing the largest ionic radii display the highest turnover frequencies and enantioselectivities in the hydroamination for these systems [75].

**Table 4 molecules-28-02702-t004:** Catalytic asymmetric hydroamination reactions of **1a**–**e** and **2c**,**d** using chiral rare earth metal complexes **15a**–**n**, **16a**–**d**, **17a**, **18** and **19a**,**b**.

Entry	Cat.	R*	[cat] [mol-%]	Substr.	Prod.	T [°C]	t [h]	Conv. [%]	*ee* [%] ^a^	Ref.
1	**15a**	*^i^*Pr	5 ^b^	**1b**	**3b**	23 ^b^	n.a.	>98	6 (*R*)	[75]
2	**15b**	*^t^*Bu	5 ^b^	**1b**	**3b**	23 ^b^	n.a.	>98	39 (*R*)	[75]
3	**15c**	*^i^*Pr	10	**1d**	**3d**	22	0.25	>99	43 ^c^	[76]
4	**15d**	Bn	10	**1d**	**3d**	22	0.25	>99	30 ^c^	[76]
5	**15e**	*^i^*Pr	10	**1b**	**3b**	30	72	>99	7 ^c^	[76]
6			10	**1d**	**3d**	22	0.25	>99	5 ^c^	[76]
7	**15f**	Bn	10	**1b**	**3b**	30	72	>99	6 ^c^	[76]
8			10	**1d**	**3d**	22	0.25	>99	6 ^c^	[76]
9	**15g**	*^i^*Pr	10	**1b**	**3b**	22	12	>99	14 ^c^	[76]
10			10	**1d**	**3d**	22	1	>99	16 ^c^	[76]
11	**15h**	Bn	10	**1b**	**3b**	22	12	>99	10 ^c^	[76]
12			10	**1d**	**3d**	22	1	>99	42 ^c^	[76]
13	**15i**	*^i^*Pr	10	**1b**	**3b**	22	168	-	-	[76]
14			10	**1d**	**3d**	22 ^d^	12	>99	36 ^c^	[76]
15	**15j**	Bn	10	**1b**	**3b**	22	168	-	-	[76]
16			10	**1d**	**3d**	22 ^d^	12	>99	46 ^c^	[76]
17	**15k**	*^i^*Pr	10	**1b**	**3b**	22	12	>99	14 ^c^	[76]
18			10	**1d**	**3d**	22 ^d^	12	>99	30 ^c^	[76]
19	**15l**	Bn	10	**1b**	**3b**	22	12	>99	12 ^c^	[76]
20			10	**1d**	**3d**	22 ^d^	12	>99	30 ^c^	[76]
21	**15m**	*^i^*Pr	10	**1d**	**3d**	22 ^e^	0.25	>99	38 ^c^	[76]
22	**15n**	Bn	10	**1d**	**3d**	22 ^e^	0.25	>99	32 ^c^	[76]
23	**16a**	-	5	**1b**	**3b**	60	5.5	95	2	[77]
24	**16b**	-	5	**1b**	**3b**	10	168	95	66	[77]
25	**16c**	-	5	**1b**	**3b**	25	288	95	5	[77]
26	**16d**	-	7	**1b**	**3b**	25	8	95	11 ^c^	[78]
27			10	**1c**	**3c**	25	0.5	100	11 ^c^	[78]
28			8	**2c**	**4c**	25	64	100	5 ^c^	[78]
29	**17a**	(+)-neomenthyl	4	**1a**	**3a**	65	65	96	22 (*R*)	[79]
30			3	**1b**	**3b**	25	12.7	96	21 (*R*)	[79]
31	**18**	(−)-menthyl	3	**1b**	**3b**	25	6.25	80	11 (*S*)	[79]
32	**19a**	*^i^*Pr	5	**1c**	**3c**	r.t.	0.17	100	34 (*S*)	[56]
33			5	**2d**	**4d**	r.t.	0.83	100	22 (*S*)	[56]
34	**19b**	*^t^*Bu	5	**1c**	**3c**	r.t.	0.17	100	93 (*S*)	[56]
35			5	**1d**	**3d**	r.t.	0.17	100	94 (*S*)	[56]
36			5	**1e**	**3e**	r.t.	3	95	89 (*S*)	[56]

^a^ Enantiomeric excesses (=*ee*) determined either by ^1^H or ^19^F NMR spectroscopy after reaction of Mosher’s acid chloride or by HPLC after naphthoylation, or determined by chiral shift ^1^H NMR spectroscopy using (*R*)-(*O*)-acetylmandelic acid to allow for the distinction between the two enantiomers. ^b^ 6 mol-% ligand. ^c^ Absolute configuration not determined. ^d^ In toluene-d^8^. ^e^ In bromobenzene-d^5^. n.a.—not applicable.

**Scheme 2 molecules-28-02702-sch002:**
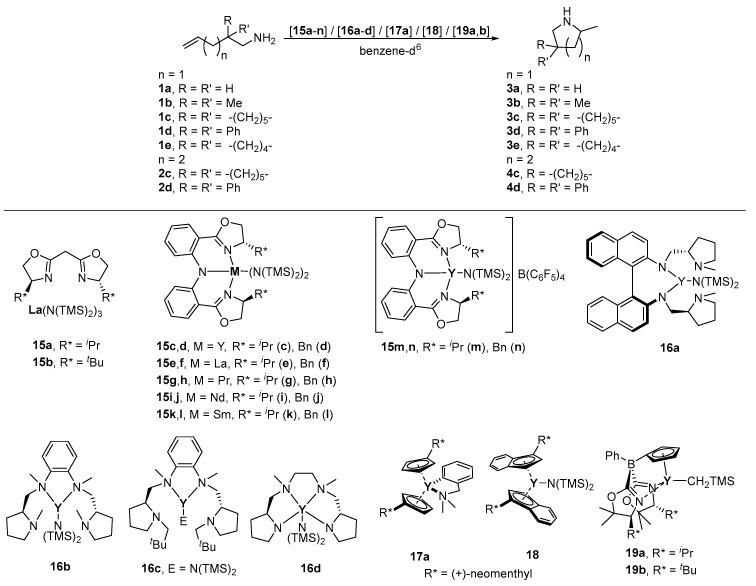
Catalytic asymmetric hydroamination of aminoalkenes **1a**–**e** and **2c**,**d** using chiral rare earth complexes **15a**–**n**, **16a**–**d**, **17a**, **18** and **19a**,**b** [56,75,78,79]. (For more details concerning catalysis data see Table 4).

Ward et al., discussed the application of bis(oxazolinylphenyl)amide(=BOPA) rare earth metal complexes **15c**–**l** (M = Y, La, Pr, Nd, Sm) in the hydroamination/cyclization of **1b**,**d** (Table 4, entries 3–20) [76]. Enantioselectivities of a maximum of 46% for catalyst **15j** (Table 4, entry 16) could be reached. Additionally, anionic yttrium catalysts **15m**,**n** were studied, showing lower enantiomeric excesses for **3d** than their respective neutrally charged counterparts **15c**,**d**.

Kim et al., reported three chiral-pool-based yttrium catalysts (**16a**–**c**) in which two alkylated (**16a**,**b**, R = Me; **16c**, R = CH_2_-tert-Bu) L-proline-derived moieties are attached to a 2,2′-diaminobinaphthyl (**16a**) or 1,2-diaminobenzene (**16a**,**b**) backbone for the intramolecular hydroamination of **1b** (Figure 2, Table 4, entries 23–25) [77]. Complexes **16a**–**c** displayed excellent activities with conversions of 95%; only **16b** showed good selectivity (66% *ee*) (Table 4, entry 4), while both **16a** and **16c** displayed only very low *ee* values [77].

In 2007, Carpentier et al., reported on the successful application of the yttrium catalyst **16d** (Figure 2; Table 4, entries 26–28), comprising a *C*_2_-symmetric chiral tetradentate diamine–diamide ligand with two L-proline-derived building blocks attached to the *N*,*N*’-dimethylethylenediamine backbone, in the intramolecular hydroamination of aminoalkenes **1b**,**c** and **2c**. Despite the high activities, only *ee* values as high as 11% could be reached for **3b**,**c** [78]. In addition, **16d** is suited for the rac-lactide ring-opening polymerization at ambient temperatures, whereby isotactic-enriched polylactides were formed [78].

The Hultzsch group published the synthesis, chemical and physical properties of (+)-neomenthyl-functionalized cyclopentadienyl and indenyl yttrocene complexes **17a** and **18** (Figure 2, Table 4 and Table 11) [79]. The synthetic methodology to prepare **17a** includes a facile arene elimination starting from [Y(*o*-C_6_H_4_CH_2_NMe_2_)_3_], while **18** was accessible by salt metathesis from the lithium species and YCl_3_. for comparison, the (−)-phenylmenthyl derivative **17b** was also prepared. Complexes **17a** and **18** displayed moderate to good catalytic activity in the tested asymmetric hydroamination reactions (Table 4, entries 9–1, and Table 11, entries 1 and 2), but only low to moderate enantioselectivities of up to 22% (Table 4, entry 29) for **17a** and 11% *ee* (Table 4, entry 31) for the sterically more hindered catalyst **18** were observed in the cyclization of **1a**,**b** [79]. The catalytic activity and enantioselectivity of non-chiral-pool-derived **17b** was comparable to **17a**. Furthermore, the authors indicated that the protolytic loss of an indenyl ligand in **18** occurs at low catalyst loading (⩽0.5 mol-%), when applying the sterically undemanding substrate **1a** [79].

In 2011, Manna et al., introduced a highly enantioselective bis(amido)yttrium complex based on chiral cyclopentadienylbis(oxazolinyl)borates(**19a**,**b**), in which the chirality is induced by L-valinol-(**19a**) and L-*tert*-leucinol-derived(**19b**) moieties (Figure 2) [56]. The catalyst **19b** in the intramolecular hydroaminations of primary aminoalkenes **1c**–**e** (Table 4, entries 12–14) and aminodialkenes **20a**–**d** (Table 11, entries 3–6) showed excellent activities and yielded the corresponding pyrrolidines with high optical purities ranging from 89% to 94% *ee* (Table 4, entries 34–36) in the synthesis of **3c**–**e** or from 92% to 96% (Table 11, entries 3–6) for the transformations of **20a**–**d**. The achieved values for the enantiomeric excess are comparable to those obtained for the isostructural zirconium complex (Tables 10 and 11) [56]. However, the (*R*)-configuration of the generated stereocenter is opposite to the pyrrolidines **3c**–**e** formed with the yttrium analog **19b**. Furthermore, the authors report on mechanistic studies, indicating that **19b** reacts by concerted C–N and C–H bond formations, which is maintained by the kinetic rate law for conversion, saturation of the respective substrate under initial rate conditions, isotopic enantioselectivity disruption and kinetic isotope effects [56]. By carrying out N–H/N–D kinetic studies, Manna et al., were able to show that the stereochemistry determining step for both Y and Zr catalysts involves an N–H (or N–D) bond. They demonstrated that the (*S*)-diastereomeric pathway is slowed down to greater extent than the *(R*)-pathway for both metal centers. Based on these results, they conclude that the catalysts have similar transition states but are of opposite energetic favorability, resulting in the observed difference in stereoselectivity [56].

Rare earth metal catalysts in which chiral-pool-modified ligands are present impose enantioselectivity, showing moderate to excellent activities in the intramolecular hydroamination for a variety of substrates. BOX-based yttrium complex **19b** displays a 96% *ee*, and shows similar activities and *ee* values in comparison to non-chiral pool catalysts of which biaryls such as BINOL or 2,2′-bis-(diphenylphosphinoamino)-1,1′-binaphthyl(=BINAM) derivatives are the best studied examples [4,5,30,80,81,82,83,84,85,86,87]. Hultzsch et al., for example, described 3,3′-bis(trisarylsilyl)- and 3,3′-bis(arylalkylsilyl)-substituted binaphtholate rare earth metal complexes (M = Y, Lu) for the hydroamination/cyclization of **1a**–**d** and **2d** with enantioselectivities of up to 95% (M = Lu) and 90% (M = Y) [88]. Overall, for the conversions, no difference is observable (>95%). On the other hand, enantioselectivities often vary significantly. For **1a**,**b**, the difference in *ee* for the yttrium catalysts is with 14% (**1a**, (*R*)-**5a**,**b**) and 6% (**1b**, **16b**) rather moderate. For **1c**,**d** the chiral-pool-based system **19b** performs with a difference in *ee* of 4% and 9%, which is better than the non-chiral-pool-derived ones. For **2d**, the respective difference in enantioselectivity is 29% (**19a**), higher in favor of the non-chiral-pool-based catalysts. For luthetium, the difference between chiral pool and non-chiral pool catalysts grows even larger: 59% for substrate **1a** ((*R/S*)-**5n**) and 49% for **1b** ((*R*)-**5m**) [88].

Chai et al., reported on a tridentate-linked amido–indenyl yttrium complex on the basis of 1,2-diaminocyclohexane, which transforms amino-olefins **1b**–**d** and **2b**–**d** into the corresponding *N*-heterocycles with *ee* values of up to 97% [89]. Those systems show a similar enantioselectivity as the 3,3′-bis(arylalkylsilyl)-substituted binaphtholate complexes towards aminoalkenes **1b**–**d** and a higher enantioselectivity towards **2d**, which increases the difference to the chiral-pool-derived catalyst even further.

#### 2.2.2. Group IV and V Metals

Aminoalcoholates of titanium (**21a**–**c**, **22a**–**c**, **23a**–**i**, **25a**–**i**, **27a**–**j** and **29a**–**h**), zirconium (**32a**–**g**, **33a**–**e**) and tantalum (**24a**–**l**, **26a**–**l** and **28a**–**j**) (Table 5, Table 6, Table 7, Table 8, Table 9, Table 10, Table 11 and Table 12), as well as the cyclopentadienylbis(oxazolinyl)borate group IV metal complexes **30a**–**c** and **31** are admirable enantioselective hydroamination/cyclization catalysts for a variety of different aminoalkenes, aminodialkenes and aminoallenes, as shown by Johnson [90,91,92,93,94,95] and Sadow [56,58,96]. Complexes **21**–**33** commonly feature natural chiral-pool-derived ligands based on either L-valine (**21a**–**c**, **23a**–**i**, **24a**–**l**, **29e**, **30a**–**c**, **32a**–**g**), L-phenylalanine (**22a**–**c**, **25a**–**i**, **26a**–**l**, **27a**–**j 28a**–**j**, **29a**–**d**), L-tert-leucine (**31**), L-proline (**33a**–**d**) and L-pipecolic acid (**33e**) (Table 5, Table 6, Table 7, Table 8, Table 9, Table 10, Table 11 and Table 12).

Chiral titanium aminoalcohol catalysts **21a**–**c** and **22a**–**c** with *N*-alkyl substituents R = 2-Ad (= 2-adamantyl), *^c^*C_6_H_11_ or *^i^*Pr allowed the effective ring-closing hydroamination of substituted aminoallenes **34a**,**b** (**34a**, Table 5; **34b**, Table 6) [90]. The cyclization of hepta-4,5-dienylamine **34a** resulted in the formation of a mixture of the six-membered 6-ethyl-2,3,4,5-tetrahydropyridine **35** (19–33% yield, Table 5) and the five-membered *Z*-(*Z*-**36**) as well as *E*-pyrrolidines (*E*-**36**) (67–86% combined yield) with *ee* values of up to 8% (*Z*-**36**) and 16% (*E*-**36**) at 110 °C (Table 5).

In contrast, the cyclization of the more sterically hindered 6-methylhepta-4,5-dienylamine **34b** afforded exclusively five-membered 2-(2-methylpropenyl)pyrrolidine **37** with high conversions (Table 6). Nevertheless, the enantiomeric excesses of **37** are with a maximum of 15% *ee* (Table 6, entry 6) [90]. A significantly higher rate acceleration when using **21a**–**c** and **22a**–**c** as catalysts in comparison to the titanium complex Ti(NMe_2_)_4_ was observed. It is still an open question if either isolated or in situ-generated metal imidos, which are common for group IV catalysts, are the catalytic active species [90]. Comparative experiments with phenylglycine-derived ligands (= Phg) were carried out showing similar activities towards **34a**,**b** as for **22a**–**c** [90].

In 2009, Johnson et al., extended the series of aminoalcohol-based titanium catalysts **21a**–**c** and **22a**–**c** towards the more bulky chiral compounds **23a**–**i** and **25a**–**i** by replacing R = H by R = methyl, *^n^*butyl or phenyl groups [91]. The corresponding ligands were prepared by a consecutive two-step synthetic procedure, whereas catalysts **23a**–**i** and **25a**–**i** were generated in situ. Intramolecular hydroamination of aminoallene **34b** exclusively results in pyrrolidine **37** with enantiomeric excesses of 16% (max.) at 135 °C (Table 6, entries 30 and 33) with quantitative conversions. No correlation between the steric bulk of the ligands and the *ee* values could be identified [91].

The Johnson group later used the previously discussed aminoalcohols (vide supra) for the preparation of the respective tantalum complexes (catalysts **24a**–**l** and **26a**–**l**) [92]. In comparison with titanium complexes **21a**–**c**, **22a**–**c, 23a**–**i** and **25a**–**i**, which are dimeric in the solid state, tantalum compounds **24a**–**l** and **26a**–**l** are monomeric possessing a somewhat distorted trigonal-bipyramidal structure as confirmed by single crystal X-ray structure analysis. Next to the chiral pool motifs derived from L-valine and L-phenylalanine, non-natural D-valine and D-phenylalanine were also studied. The best results in the cyclization of aminoallene **34b** to pyrrolidine **37** were obtained by catalysts containing R’ = Ph as substituents (**24d**,**h**,**l** and **26d**,**h**,**l**). Enantioselectivities ≤ 80% *ee* were obtained with a 5 mol-% catalyst loading (Table 6, entries 19, 23, 27, 40, 44 and 48) [92]. Generally, the tantalum derivatives show better *ee* values than those of the respective titanium catalysts at the cost of higher reaction times and a greater variance in conversion rates.

Crowded sulfonamides featuring a benzyl group as a bulky chiral backbone (L-phenylalanine-derived) with different steric and electronic properties were successfully introduced as ligands for the in situ generation of titanium (**27a**–**j**) and tantalum (**28a**–**j**) catalysts [93,95]. The respective titanium catalysts convert 6-methylhepta-4,5-dienylamine **34b** solely to 2-(2-methylpropenyl)pyrrolidine **37** with an enantiomeric excesses of max. 11% (**27a**–**f**) (Table 7, entries 1–12) [93] or 18–41% (**27g**–**j**) (Table 7, entries 13–17) [95] with conversions of 18–100%. The corresponding tantalum catalysts generally showed an *ee* of 5–34% (**28a**–**f**) (Table 7, entries 18–29) [93] and 33–39% (**28g**–**j**) (Table 7, entries 30–33) [95] more selective with generally higher conversions than **27g**–**j**.

In the hydroamination/cyclization of hepta-4,5-dienylamine **34a** using **27g**–**j** and **28g**–**j** as catalysts, a mixture of tetrahydropyridine **35** (48–82% yield) and *Z*-**36a** and *E*-**36b** (15–50% combined yield) was obtained with *ee* values of up to 55% (*E*-**36a**) and 45% (*Z*-**36b**) (Table 8, entry 3) with tantalum showing an higher pyrrolidine yield and reduced enantioselectivities.

A further modification of the earlier discussed titanium catalysts **21a**–**c** and **22a**–**c** (Table 5 and Table 6, entries 1–6), which are suitable for aminoallene ring-closing reactions, was carried out by the introduction of chiral, tridentate, dianionic imine-diol ligands at the titanium metal center, resulting in the formation of **29a**–**h** (**29a**–**e**, Table 8; **29a**–**h**, Table 9) [94,97]. Nevertheless, cyclization of hepta-4,5-dienylamine (**34a**) resulted in a mixture of tetrahydropyridine **35** (40–72% yield) and pyrrolidines *Z*-**36** (8–17% yield) and *E*-**36** (17–39% yield) (Table 8). Using **34b** as a substrate, **37** was exclusively produced in the presence of **29a**–**h** (Table 9) as already described for the catalytic systems **21a**–**c** and **22a**–**c** (Table 6). The *ee* values of a maximum of 22% are comparable to the values observed for **21a**–**c** and **22a**–**c** with comparable conversions (Table 9, entry 10) [90,97].

For the intramolecular hydroamination of aminoalkanes using chiral-pool-derived catalysts, Sadow et al., published the highly enantioselective bis(amido)zirconium complex **30b** possessing a chiral cyclopentadienylbis(oxazolinyl)borate in which chirality is induced by the incorporation of L-valinol into the ligand (Figure 3) [56,57,96]. The addition of catalytic amounts of **30b** to primary aminoalkenes **1a**–**f**, **2c**,**d** and **38**–**40** yielded the corresponding *N*-heterocycles **3a**–**f**, **4c**,**d** and **41**–**43** with enantiomeric excesses ranging from 31% for **2c** (Table 10; entry 22) to 98% for **1d** (Table 10; entries 14; 18). It was proposed that the observed reactivity and high enantioselectivity of **30b** may relate to the ability of the relevant intermediate to stabilize the proposed six-center transition state [56,57]. Curiously, complex **30b** and its yttrium derivative **18** (vide supra) gave pyrrolidines **3c**–**e** and **44d** with an opposite absolute configuration, despite using the same ligand system as the (*R*)-derivative of **30b** using D-valine as chiral building block. In addition, the L-tert-leucine derivative **31** was prepared in a multiple-step synthetic procedure [56]. The catalytic performance of **31** on the cyclization of aminoalkenes **1c**–**e** and **2c** corresponds to L-valine-derived **30b**, resulting in similar conversions with a maximum of 93% (Table 10; entry 36) and 29% *ee* (Table 10; entry 38) with generally high conversions. The existence of a kinetic rate dependence was further shown, evolving from a 1st order at a low substrate concentration to zero-order at a high concentration, which is representative of a reversible catalyst/substrate interaction preceding the N–H bond cleavage in the turnover-limiting and irreversible step of the catalytic cycle [56].

Exchanging zirconium in **30b** by titanium (**30a**) or hafnium (**30c**), the latter two species catalyze the cyclization of amino-olefins **1b**–**f**, **2c**,**d** and **38**–**40** to result in the corresponding *N*-heterocycles **3b**–**f**, **4c**,**d** and **41**–**43** in enantiomeric excesses of 76–82% (**30a**) or 18–97% (**30c**) with moderate to high conversions (for more details see Table 10) [58]. This work was extended to aminodialkenes **20a**–**h** and aminodialkynes **45a**–**c** using **19b**, **30b**,**c** and **31** as catalysts as depicted in Figure 4 [56,57,58,96]. Diastereomers **44a**–**h** (Figure 4) of five- to seven-membered *N*-heterocycles were obtained when aminodialkenes **20a**–**h** were used as substrates, while in the case of aminodialkynes **45a**–**c,** the respective imines **46a**–**c** were produced in an enantioselective reaction in high to moderate yields. Depending on the cyclization conditions applied, diastereo- and enantioselectivities of max. 99% could be reached using zirconium catalyst **30b** (Table 11, entries 13–15) [96]. In comparison, yttrium-based systems **17a** and **18** reached lower *ee* values of up to 38% (Table 11, entries 1 and 2), while **19b** showed similar enantioselectivities to **30b**. It was found that catalytically generated stereocenters in cyclized **44a**–**h** can be independently controlled by the catalyst’s properties and reaction conditions (Table 11). At low concentrations *Z*-**44b** is favored, and at high concentrations combined with lower temperatures, *E*-**44b** (Table 11, entries 7–9) is favored. It could be further demonstrated that isotopic substitution of hydrogen by deuterium (H_2_NR/D_2_NR in **20b**) significantly improved the diastereoselectivity from the ratio of 8:1 to a maximum of 43:1 and increased the optical purity to 99% *ee* [96]. As demonstrated for **30b**, experimental studies on aminodialkene ring-closing reactions to examine the effects of the catalyst-to-substrate ratio, the absolute catalyst concentration and the absolute original substrate concentration show that the latter parameter greatly influences the stereoselectivity, whereas the absolute configuration of the *α*-amino stereocenter created by the C–N bond generation is not influenced by any parameters of the concerted proton-triggered cyclization mechanism (Figure 3) [96]. Coordination of a primary amine changes the ring conformation in the transition state to place the *cis* group axial to avoid unfavorable interactions between the bulkier substituent and the cyclizing substrate resulting in the formation of the *trans* diastereomer. With decreasing concentrations of the primary amine, pathway **B** becomes more unlikely, while cycle **A** is more favored, resulting in the increased formation of the *cis* diastereomer [96]. With amine deuteration, the coordination becomes more hindered, resulting in the observed increase in the respective *cis* product.

Furthermore, dibenzyl zirconium complexes **32a**–**g** and **33a**–**e** (Figure 5, Table 12) have been applied in the cyclization of primary aminoalkenes **1b**–**e**, **2d** and **20a**,**b** [98]. The chirality of the appropriate catalyst was introduced by the L-valine- (**32a**–**g**), L-proline- (**33a**–**d**) or L-pipecolic acid-derived (**33e**) backbone of the tridentate dianionic amino–diol ligand with variation possibilities being at the ether functionality (**32a**–**g**) or the substituents of the *α*-position to the alcohol functionality (**33a**–**e**) and the aromatic substituent R’ in the ligand system (Figure 5). These catalysts show satisfactory catalytic activities in the C–N bond formation of aminopentenes **1b**–**e** and aminohexene **2b**. Conversions as high as 97% and high enantiomeric excesses (**32d**, max. 56% *ee* for **3d** (Table 12, entry 4); **33b**, up to 94% for **3d** (Table 12, entries 15 and 16)) were observed in the catalytic synthesis of five-membered pyrrolidines **3b**–**e** and *E*/*Z*-**44a**,**b** [98]. The authors also proposed a mechanism involving a highly ordered transition state and a concerted bond formation pathway. Variations in the temperature for the hydroamination of **1d** using **33b** as catalyst resulted only in minor changes in conversion and *ee* values (Table 12).

Overall, group IV metal catalysts **30a**–**c** and **33a**–**e** show high quantitative conversions with enatiomeric excesses as high as 98% for aminoalkenes **1a**–**f**, **2c**,**d** and **38**–**40** and up to 99% for aminodialkenes **20a**–**h** and aminodialkynes **45a**–**c**. Both values are comparable for aminohexene substrates **2c**,**d** and better for aminopentenes **1a**–**f**, **38**, **39** than those obtained by non-chiral-pool-derived catalysts which are mainly based on bisaryl-derived or salen-type ligands [81,84,99,100]. Comparisons of the hydroamination of aminoallenes **34a**,**b** using Ti and Ta catalysts **21a**–**c**–**29a**–**h** with those applying non-chiral-pool-derived catalytic systems, which are mainly based on bisaryl-derived ligands, are more complicated due to differences in substrate screenings [86,101,102,103].

### 2.3. Early Main-Group Elements

In contrast to transition metals, which can appear in different oxidation states defining their reactivity by *d*-electrons, early main-group elements of group I and II are primarily characterized by mono-(alkaline) or dicationic (alkaline earth) ions depending on their outer shell *s*-electrons. Hence, main-group elements cannot easily switch between oxidation states, and therefore, catalysis with those metals is solely based on polar reaction mechanisms and Lewis-acid activations [104].

#### 2.3.1. Alkaline Metals

Group I elements can be used as pre-catalysts both in their elementary as well as ionic form [8,105,106,107,108]. In enantioselective hydroamination reactions, solely lithium-based catalysts have been reported (intermolecular: **47**, **48**, Figure 6, Table 13; **49a**; **50a**–**f**, Table 14; intramolecular: **49a**,**b**, Table 15) [109,110,111,112,113,114].

Ates et al., first described the suitability of *^n^*BuLi (16 mol-%) in the catalytic high-yield synthesis of five- and six-membered *N*-heterocycles via the intramolecular hydroamination of non-activated aminoalkenes such as **1a**–**c** [105]. Shortly after, Hultzsch et al., reported the first Li-catalyzed enantioselective ring-closing reaction of 2,2-substituted pent-4-en-1-amines **1b**–**d**, **20a** and **51** (Figure 6, Table 13), providing the corresponding pyrrolidine derivatives **3b**–**d**, **44a** and **52** [110]. As a catalyst, they used the dimeric, tetranuclear (*S*,*S*,*S*)-*N,N’*-dimethylpyrrolidinediamidobinaphthyl dilithium complex **47** (Figure 6). The catalytic reactions succeeded with almost quantitative conversions and an enantiomeric excess of a max. 75% (Table 13, entry 5). The binaphthyl-centered chelate ligand in **47** is based on a DABN backbone to which two L-proline-derived moieties are attached. In the solid state, each of the four lithium ions possess different coordination environments, which exhibit a similar structure in solution [110]. It was found that only minor differences in enantiomeric excesses exist by various catalyst loadings and/or by the addition of coordinating solvents such as tetrahydrofuran. In contrast, these variations influenced the formation of the *N*-heterocyclic molecules more significantly. When, instead of the DABN backbone in **47**, naphthyl was introduced in chiral **48**, almost no enantiomeric excess and significant lower conversions in the formation of the respective *N*-heterocyclic compounds **3b**,**d** was observed. No enantioselectivities were obtained by using the combination (−)-sparteine/LiN(SiMe_3_)_2_ (**49a**) as a pre-catalyst, albeit the observed conversion of 98% is comparable to **47** [110].

In 2007, Tomioka and his group discussed the intramolecular hydroamination of aminoalkenes **53a**,**b** at −60 °C in toluene by applying in situ-produced catalytic systems **50a**–**f**, containing diverse chiral bisoxazoline (= BOX) ligands (Table 14) [113]. In kinetically controlled catalytic reactions, almost quantitative yields and good *ee* values for the formation of *N*-heterocycles **54a**,**b** was observed (Table 14). Within the catalytic active system, diisopropylamine acts as coordinating and proton-donating reagent. Out of the nine studied catalysts, **50a**–**e** contain amino acid-based chiral pool ligands, of which **50d** converted substrate **53a** into the corresponding six-membered tetrahydroisoquinoline **54a** with 84% *ee* (Table 14, entry 11), while catalysts **50a**,**c** produced five-membered isoindoline **54b** with high regioselectivity and an *ee* of 84% (Table 14, entries 2 and 10) using **54b** as substrate. In none of the cases the formation of *endo*-cyclized **55** was observed. On the other hand, the more rigid terpene camphor-modified catalyst **50f** resulted in lower activities and *ee* values for the cyclization of aminopentene **53b**, while for aminohexene **53a** comparable results to **50d** could be reached. However, both synthesized *N*-heterocycles **54a**,**b** using **50f** as catalyst possess the (*R*)-configuration instead of the (*S*)-products favored by **50a**–**e**. The best catalytic performance for the hydroamination of **53b** was found for **50g** having the non-chiral-pool D-isoleucine-derived groups attached to the BOX ligand (91% *ee*). Exchanging the solvent from toluene to tetrahydrofuran for catalysis resulted in the formation of both **54a**,**b** as the kinetic and endo-cyclized **55a**,**b** as the thermodynamic product [113].

The pre-catalyst **50a** was selected for intramolecular hydroamination screening of aminoalkenes **53c**–**f** (Table 14, entries 9–16) and **56**, of which the synthesis of (*S*)-laudanosine (**57**) is exemplarily shown in Figure 7 [114]. Based on these studies, Yamamoto et al., synthesized (−)-javaberine A and (−)-epi-javaberine in an asymmetric total synthetic methodology with 76% *ee* using **50a** as catalyst (Figure 7) [115].

Catalyst **49a** (vide supra) can be successfully used in the intermolecular hydroamination of olefins **58a**,**b** with amines **59a**,**b** resulting in *ee* values of up to 14% (Table 15, entry 14) and conversions from 38–71% [109], which contrasts the earlier discussed intramolecular hydroamination reactions showing no enantioselectivity. No enantiomeric excess was observed for **49b** with (−)-*α*-isosparteine as ligand [109].

Outside of the chiral pool, Deschamp et al., reported a non-chiral-pool-based diamidobinaphthyl building block, allowing the variation in alkyl and methylene-aryl substituents at the amino functionalities [111,112]. Addition of LiCH_2_SiMe_3_ to the respective *N*,*N’*-disubstituted binaphthyldiamine resulted in the corresponding in situ-generated chiral lithium catalysts. Their use in the cyclization of conjugated 1,3-aminodienes **11** and **12a** results in **13** and **14a** with *E*/Z selectivities and *ee* values of up to 72%, while aminopentenes including **1b**–**d**, **2c** and **51** provided **3b**–**d**, **4c** and **52** with a maximum of 58% *ee* [111,112]. The enantioselectivities of the latter catalysts are for **1b** (Δ*ee* = −61%), **1c** (Δ*ee* = −63%) and **51** (Δ*ee* = −15%), significantly lower, and for **1d** (Δ*ee* = 27%), higher, than for **47** [109]. To the best of our knowledge, no other chiral lithium catalysts were so far reported for the discussed substrates.

#### 2.3.2. Alkaline Earth Metals

The first alkaline-earth-metal-mediated hydroamination catalysis was achieved by Hill et al., in 2005 using achiral heteroleptic *β*-diketiminato calcium complexes [6,34].

In 2009, Hultzsch et al., published the first chiral-pool-based alkaline earth magnesium catalysts (*S*,*S*,*S*)-**61** and (*R,S*,*S*)-**61** for the cyclization of aminoalkenes **1b**–**d** (Figure 8, Table 16) [116]. In contrast to the lithium derivative **47** (vide supra), magnesium complexes (*S*,*S*,*S*)-**61** and (*R*,*S*,*S*)-**61** with their L-proline-derived axial chiral tetraamine ligands show moderate to high catalytic activities, but only limited enantiomeric excesses, with a maximum of 14% (Table 16, entry 6), due to the protolytic ligand exchange processes as typical for heteroleptic alkaline earth metal complexes. This solution-based phenomenon is known as the Schlenk equilibrium [117,118,119]. Within reference [116], the zinc derivatives of (*S*,*S*,*S*)-**61** and (*R,S*,*S*)-**61** were prepared. It was found that they are active hydroamination catalysts, yielding higher *ee* values (up to 29%) as their magnesium homologs [116].

In 2011, Sadow et al., described the synthesis of the magnesium complex **62a** comprising a chiral, pseudo *C*_3_-symmetric, mono-anionic tris(oxazolinyl)borato ligand (Figure 8) [120]. Its use in hydroamination reactions was also reported. The chirality of **62a** results from three L-*tert*-leucine moieties. Due to the bulkiness of the respective ligand, the Schlenk equilibrium is hindered. Catalyst **62a** produced good to excellent conversions in the intramolecular hydroamination of **1b**–**d** (Table 16, entries 7–12). The enantiomeric excesses, as compared with structurally similar complexes **31** and **32** (*vide supra*), were with a max. of 36% *ee* lower [120].

Overall, complexes (*S*,*S*,*S*)-**61**, (*R*,*S*,*S*)-**61** and **62a**, with their chiral-pool-derived motifs, show similar activities and conversions in the intramolecular hydroamination of **1b**–**d** as (*R*,*R*)-[{ONN}MgCH_2_Ph] (**63**, ONN = (*R,R*)-*tert*-butyl-2-(((-2-(dimethylamino)-cyclohexyl)(methyl)amino)methyl)-6-(triphenylsilyl)phenolato) [121], however, their enantiomeric excess is considerably lower (**63**, up to 90% *ee* for **3b**–**d**). Mechanistic studies on **63** were carried out comparing an *σ*-insertive mechanism against a concerted non-insertive one. DFT studies confirmed that proton-assisted concerted C–H/C–N bond formation is energetically not favored, contrary to the kinetically less demanding *σ*-insertive path [29,68,121]. The observed *ee* values for chiral-pool-derived magnesium-based catalysts are overall lower than those for non-chiral pool-derived **63**.

In addition to **62a**, the isostructural optically active calcium complex **62b** was synthesized (Figure 8) [120]. It was observed that within this species the Schlenk equilibrium is hindered in solution, as evidenced by NMR and IR studies. Catalyst **62b** showed increased activities and quantitative conversions after minutes in comparison to **62a**, but the stereoselectivity decreased to 18% *ee* for **3b**,**c** (Table 16, entries 13 and 14) [120].

The first chiral-pool-derived (L-valine) calcium catalysts **64a**–**d** (Figure 9) for enantioselective hydroamination reactions of aminoalkenes **1b**,**d** originate from the Ward group, showing for **64a**,**b** (Table 17, entries 1–4) similar activities and conversions (> 90%) when compared to **62b** (Table 16, entries 13 and 14) [26]. Nevertheless, only an enantiomeric excess of 0–12% was observed for **3b**,**d**. It should be mentioned that the para-fluorophenyl derivative **64d** displayed no activity for substrates **1b**,**d**, even after several weeks. Catalyst **64c** (R = Ph) also revealed no activity when using **1b** as substrate, while in the cyclization of **1d** an 80% conversion occurred with 26% *ee* (Table 17, entry 6) [26]. This enantioselectivity signifies a notable increase in *ee* as compared with **64a**,**b** (vide supra). It is also higher than the values reported for the calcium complex **62b** and other non-chiral-pool-based BOX-containing calcium systems published by Buch and Harder [117,120].

In 2011, Wixey and Ward described the use of chiral-pool-based bisimidazoline calcium complexes **65a**–**c** in the catalytic cyclization of aminoalkenes **1b**,**d** (Figure 9) [122]. Like **64a**–**d**, complexes **65a**–**c** are derived from L-valine as a chirality inducing motif. It was shown that the ligand redistribution through the Schlenk equilibrium depends on the substituents R [122]. The measured *ee* values are within <12% low, however, they compare well to those for **64a**–**c** and the complexes containing other non-chiral-pool-derived BOX ligands [117,120].

In 2012, Nixon and Ward extended the series of bisoxazoline calcium complexes by bis(oxazolinylphenyl)amines(=BOPA), of which two of the three introduced BOPA-based catalysts (**66a**,**b**) (Figure 9) derive from the chiral pool (L-valinol, L-phenylalaninol) [123]. In the enantioselective hydroamination of **1b**, quantitative conversions and *ee* values of up to 26% *ee* could be achieved (Table 17, entry 22). The conversion for aminohexene **2d** was determined to be 0–83% with enantiomeric excesses as high as 16% at 80 °C (Table 17, entry 26). A major improvement in stereoselectivity (as high as 50% *ee* for **1d**) could be reached by employing BOPA ligands based on the non-natural, non-protogenic amino acid L-*α*-phenylglycine [123]. This significant improvement is due to the relatively slow ligand redistribution rate. A further increase in enantioselectivity to 56% *ee* for substrate **1d** was reported by Harder et al., using non-chiral pool BINAM derivatives as bulky dianionic ligands [28].

While the use of free alcohols as ligands is rather common for early transition metals such as titanium or tantalum, their application in alkaline-earth-metal-based catalysts is rather limited, with phenoxyamine **63** from the Hultzsch group being the most prominent one in the case of magnesium [121]. However, no system is currently used which incorporates structural motifs derived from the chiral pool. In the case of calcium, alcoholates have, up to now, not been used at all. Therefore, we expanded on this type of binding site with the synthesis of an amino acid-derived tertiary alcohol (Figure 10). This tridentate proto-ligand is accessible from L-isoleucine via a cascade of reductive aminations followed by a Grignard reaction. The transformation of aminoalkene **1d** to the respective pyrrolidine **3d** in a yield of >99% with an enantiomeric excess of 67% could be obtained by in situ formation of the catalyst **67** (Figure 10) [124]. To the best of our knowledge, this enantioselectivity is the highest observed one for calcium-based species, including non-chiral-pool-derived catalysts, which greatly illustrates the potential of such compounds in the area of intramolecular hydroamination reactions.

Another type of catalysts for the enantioselective intramolecular hydroamination is based on the application of alkaline earth metals as pure Lewis-acidic metal centers. For example, non-basic calcium iodide as a Lewis acid and an external base for deprotonation can be used [27,125]. In general, it was found that the activities of alkaline earth metal iodides decrease in the series Ca > Sr >> Mg > Ba [27]. The proposed mechanism is shown in Figure 4. Coordination of the amino alkene to CaI_2_ acidifies one of the two NH_2_ protons. Deprotonation occurs by the ^tBu^P4 phosphazene base, followed by the cyclization of the amino-olefin at the calcium metal center. After protonation of the formed *N*-heterocycle by [*^t^*^Bu^P4H]^+^, pyrrolidine is released [125]. As chiral catalysts, (−)-fenchone-based **68** and non-chiral-pool-based BINOL-modified **69** were applied (Table 18). Catalyst **68** along with *^t^*^Bu^P4 gave in the enantioselective hydroamination of aminoalkenes **1b**–**d** pyrrolidines **3b**–**d** with almost quantitative conversions and *ee* values reaching 15% (Table 18, entries 3 and 4). The experimentally determined *ee* values are generally lower than those for (*S*)-**69**, which achieves enantioselectivities of a max. 33% (Table 18, entries 7 and 8) [27].

In general, amido or benzyl strontium and barium complexes are also active in hydroamination reactions. Their overall activity is, however, lower than that of calcium, and no chiral catalyst based on the natural chiral pool have yet been reported [69,118,126,127].

## 3. Conclusions

The hydroamination reaction is an atom economical possibility for C–N bond formation, starting from common functional groups such as an amino functionality together with unsaturated *C,C* bonds. One of the main challenges arises from the high reaction barrier, which is attributed to the strong electronic repulsion of the participating groups and the symmetry forbidden nature of the [2 + 2] cyclization [5,6]. Hence, the application of catalysts is required. Over recent years, a vast amount of different catalysts were investigated, not only allowing for a maximum in yield, but also to ensure the stereoselectivity required for such transformations in case of asymmetric reaction products. Alkaline (Li), alkaline earth (Mg, Ca), rare earth (Y, La, Nd, Sm, Lu), group IV (Ti, Zr, Hf) metals, and tantalum are heavily applied in this field of research. With the rising demand for cheap and easily accessible catalysts, a promising strategy for the induction of chirality is the use of moieties obtainable from the chiral pool. In this case, the majority of ligand systems is derived from amino acids, while terpenes and alkaloids are only applied scarcely.

The chiral-pool-based building blocks can be incorporated into the ligands by different strategies, with the most prominent motifs being alcoholates, ethers and (bis)oxazolines. The best performing systems for titanium (**27i**, Phe-derived, Table 7), tantalum (**24h**, Phe-derived, Table 6) and calcium (**67**, Ile-derived, Figure 10) are based on bulky amino-alcohols. For magnesium (**62a**, Val-derived, Table 16), yttrium (**19b**, Tle-derived, Table 4 and Table 11) and zirconium (**30b**, Val-derived, Table 10) BOX-containing ligands showed the best results, while for lanthanides, (−)-menthyl-substituted *ansa*-complexes performed well in the intramolecular hydroamination of aminoalkenes **1a**–**f** and **2a**–**d**.

The resulting complexes are often of equal reactivity and selectivity than their non-chiral-pool-based, often bisaryl-derived counterparts. Therefore, they are a good alternative to established catalytic systems, with the exception of magnesium catalysts, which show significantly lower enantioselectivity compared to non-chiral-pool-derived ones, such as the phenoxyamine-based system **63**. However, comparison between chiral-pool- and non-chiral-pool-derived titanium and tantalum catalysts is complicated due to the differences in substrate screening and the low amount of chiral catalytic systems found in the literature [101,102].

While a variety of different substrates, such as aminoalkenes, aminodialkenes and aminoallenes are investigated with great success, applications on higher functionalized substrates are only viewed scarcely and are often limited to a narrow number of model systems.

In summary, a range of different catalysts based on early metals is nowadays available for the application in hydroamination reactions, greatly enhanced by motifs originating from the chiral pool. Progress has been made towards high performant systems accompanied by a detailed understanding of their reaction behavior. Today, those catalysts are comparable to their more expensive heavy-transition-metal-based counterparts, often using cheaper and more accessible ligand systems. Their main limitation resides in the scarce substrate scope against which those catalysts were tested, greatly diminishing the possibilities arising from those catalysts. Therefore, upcoming challenges for early-metal-based hydroamination reactions need to shift from a pure catalyst development stage towards applications on more complex targets relevant for, e.g., pharmaceuticals or fine chemicals. By doing so, the extensive knowledge on early-metal-based catalysts can be harnessed and tailored to further enhance the toolkit in organic chemistry towards more (atom)economic and sustainable synthetic routes.

## Data Availability

Not applicable.

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
