# Peer review of "Jumping in the Chiral Pool: Asymmetric Hydroaminations with Early Metals"

_molecules, 2023, doi:10.3390/molecules28062702_

Round 1

Reviewer 1 Report

In this manuscript (Jumping in the Chiral Pool: Asymmetric Hydroaminations with Early Metals), the authors reviewed the literature on the well-known, important in organic chemistry, asymmetric hydroamination reaction. The high energy barrier of activation in the reaction requires the use of catalysts, preferably complexes of cheap metals. The research of many chemists is focused on achieving the best relationship between stereo-selectivity and final yield. It is usually very difficult to achieve both parameters in one reaction and requires many well-planned experiments. The review publication is interesting since it presents quite interesting approach to catalysis. The text is good written and the discussion contains a lot of details in order to understand the main idea.

In summary: I think, this article review sounds well-written and organized and it could be published in Molecules.

Author Response

Response to reviewers #1

In summary: I think, this article review sounds well-written and organized and it could be published in Molecules.

Perfect. Thanks to the reviewer!

Reviewer 2 Report

This is a valuable and well structured and presented overview of major advances in the development of enantioselective hydroamination catalysis. The introduction provides a clear and historically accurate context for the main body of discussion, which focuses on the development of systems derived from the more electrophilic rare earth, early transition metal and s-block elements.

The subsequent review is very well organised and appropriately referenced with major advances represented in clearly drawn schemes and figures, and with the primary data exemplified in a series of quite detailed tables. The review is also very nicely written and easy to digest, such that I am convinced of its value, despite the relatively high profile of the subject matter, which has been the subject of earlier, though more focused reviews.

In short, therefore, I would be be very much in favour of acceptance of this submission, effectively as submitted. 

Author Response

Response to reviewers #2

In short, therefore, I would be be very much in favour of acceptance of this submission, effectively as submitted.

Many thanks to the reviewer.

Reviewer 3 Report

Recommendation: Accept with minor revision

Comments:
The claim of the review is "Jumping in the Chiral Pool: Asymmetric Hydroaminations with

Early Metals" by Heinrich Lang, which is a well-chosen topic in present scenario. I would like to recommend this work to ‘Molecules’ after the following minor corrections. 

1. In conclusion section please write future outlook of Asymmetric Hydroaminations with Early Metals.

2. Please check whole manuscript carefully, font sizes are different, e.g. page no. 589, 623, 654, 721… etc.

Author Response

Response to reviewer #3

1) Check the whole manuscript for different font sizes (e. g. 589, 623, 654, 721).

The manuscript was checked for formatting inconsistencies originating from the transfer to the template.

2) In conclusion section, please write future outlook of asymmetric hydroamination with early metals.

As requested the conclusion was extended by an outlook section. The changed text passage at the end of the conclusion now reads as follows: “In summary, a range of different catalysts based on early metals is nowadays available for the application in hydroamination reactions, greatly enhanced by motifs originating from the chiral pool. Progress has been made towards high performant systems accompanied by a detailed understanding on their reaction behavior. Today those catalysts are comparable to their more expensive heavy transition metal-based counterparts often using cheaper and better accessible ligand systems. Their main limitation resides in the scarce substrate scope, on which those catalysts were tested against, greatly diminishing the possibilities arising from those catalysts. Therefore, upcoming challenges for early metal-based hydroamination reactions need to shift from a pure catalyst development stage towards applications on more complex targets relevant for, e. g. pharmaceuticals or fine chemicals. By doing so, the extensive knowledge on early metal-based catalysts can be harnessed and tailored to further enhance the toolkit in organic chemistry towards more (atom)economic and sustainable synthetic routes. “